# Integration Linkage Mapping and Comparative Transcriptome Analysis to Dissect the Genetic Basis of Rice Salt Tolerance Associated with the Germination Stage

**DOI:** 10.3390/ijms251910376

**Published:** 2024-09-26

**Authors:** Leiyue Geng, Tuo Zou, Wei Zhang, Shuo Wang, Yutao Yao, Zhenyu Zheng, Qi Du, Longzhi Han

**Affiliations:** 1Institute of Coastal Agriculture, Hebei Academy of Agriculture and Forestry Sciences, Tangshan 063299, China; lowrygeng@163.com (L.G.); zoutuo520@126.com (T.Z.); nkybhszw@163.com (W.Z.); wangshuo2271@126.com (S.W.); yutao890310@163.com (Y.Y.); bhsdln@163.com (Z.Z.); 2Tangshan Key Laboratory of Rice Breeding, Tangshan 063299, China; 3Institute of Crop Sciences, Chinese Academy of Agricultural Sciences, Beijing 100081, China

**Keywords:** rice, salt tolerance, germination stage, linkage mapping, RNA-seq, candidate genes

## Abstract

Soil salinity poses a serious threat to rice production. The salt tolerance of rice at the germination stage is one of the major determinants of stable stand establishment, which is very important for direct seeding in saline soil. The complexity and polygenic nature of salt tolerance have limited the efficiency of discovering and cloning key genes in rice. In this study, an RIL population with an ultra-high-density genetic map was employed to investigate the salt-tolerant genetic basis in rice, and a total of 20 QTLs were detected, including a major and stable QTL (*qRCL3-1*). Subsequently, salt-specific DEGs from a comparative transcriptome analysis were overlaid onto annotated genes located on a stable QTL interval, and eight putative candidate genes were further identified. Finally, from the sequence alignment and variant analysis, *OsCam1-1* was confirmed to be the most promising candidate gene for regulating salinity tolerance in rice. This study provides important information for elucidating the genetic and molecular basis of rice salt tolerance at the germination stage, and the genes detected here will be useful for improvements in rice salt tolerance.

## 1. Introduction

Rice is a staple crop and is grown in a wide range of ecological environments [1]. Salinization is one of the most serious abiotic stresses impeding rice productivity worldwide [2]. Approximately 30% of the total rice-growing area in the world is affected by salinity stress [3]. Recently, the direct seeding cultivation of rice has become increasingly popular, with the advantages of labor reduction and cost efficiency [4]. However, salt sensitivity during seed germination often leads to a low emergence rate and translates to reduced stand density in salt-affected paddies [5]. Therefore, good capabilities for seed germination and seedling establishment are also an essential objective for salt-tolerant rice breeding.

The vast genetic variability in germplasm in response to salt stress makes it possible to improve salt tolerance in rice [6]. However, salt tolerance is a polygenic trait that is highly influenced by environmental factors, which has presented challenges in making significant progress by conventional breeding [7]. Meanwhile, negative characteristics accompanying salt tolerance in landrace and wild cultivars act as a bottleneck in conventional breeding strategies, and this has led to increased interest in molecular breeding methods [8]. Modern molecular breeding strategies enable improved salinity tolerance by precisely transferring genes into popular varieties and pyramiding multiple relevant QTLs (quantitative trait loci)/genes for a particular stress-prone environment [9]. However, it is a prerequisite to discover QTLs conferring salt tolerance in rice followed by critical gene cloning [10]. Over the past two decades, QTL mapping for salt tolerance in rice has advanced significantly, and over a thousand QTLs have been identified as being associated with different rice developmental stages [11]. Nevertheless, most of those QTLs are primarily associated with the seedling or reproductive stage, and only rare genes have been characterized by map-based cloning or utilized in molecular breeding [12]. For instance, *SKC1* [13,14], *DST* [15], *HST* [16], *GS3* [17], and *STG5* [18] are well-known crucial halotolerant genes in rice, and the fundamental knowledge about these genes has laid the foundation for elucidating salt-tolerant genetic mechanisms in crops [19].

Rice responds differently to salt stress from germination to senescence, and numerous regulatory mechanisms have evolved [20]. Studies are still needed to identify novel main-effect QTLs or genes associated with different rice developmental stages [21]. Nevertheless, only *qSE3*, which is isolated by map-based cloning and promotes seed germination and seedling establishment under salt stress in rice, has been reported. *qSE3* encodes a K^+^ transporter gene, *OsHAK21*, which significantly increases K^+^ and Na^+^ uptake, activates ABA signaling responses, and significantly decreases the H_2_O_2_ level. All of these seed physiological changes contribute to seed germination and seedling establishment under salinity stress [22]. As compared to the seeding stage, less progress has been made in deciphering the salinity tolerance mechanism at the germination stage in rice. It is still imperative to conduct further research and uncover salt-tolerant QTLs/genes associated with the germination stage.

The traditional linkage mapping strategy does not usually provide an accurate way to distinguish candidate genes underlying the QTLs without further fine-mapping [23]. The positional cloning of QTLs may be an effective approach for the identification of genes underlying target QTLs, but this is still laborious and time-consuming [24]. In comparison to traditional QTL mapping techniques, the integration of linkage mapping and RNA-seq can expedite the identification of candidate genes conferring complex quantitative traits [25]. Utilizing the joint analysis strategy, many candidate genes affecting complex quantitative traits have been successfully identified and cloned [26]. For example, with the strategy of combined QTL mapping and transcriptome analysis, Li et al. [27] successfully revealed *LOC_Os12g29400* as a causal salt tolerance gene expressed at the rice seeding stage. Using a combination of linkage mapping and transcriptome analysis, Kong et al. [28] revealed *LOC_Os01g04430* and *LOC_Os01g04530* as promising candidate genes controlling the coleoptile length in rice. These studies demonstrated that DEG analysis within QTLs can effectively locate candidate genes.

In this study, an RIL population with an ultra-high-density genetic map was employed for exploring the genetic basis of salinity tolerance in rice. Subsequently, a supplementary comparative transcriptome analysis was applied to narrow down the range of candidate genes. Ultimately, a sequence variation analysis was utilized to pinpoint the causal gene and complete a preliminary functional verification of salt-tolerant mechanisms in rice.

## 2. Results

### 2.1. Phenotypic Variation in the RIL Population

The relative germination index (RGI), germination rate (RGR), and coleoptile length (RCL) of two parents and their derived population were evaluated under two salt conditions (150 mM and 300 mM). Compared to Milyang23, Jileng1 showed relatively higher salt tolerance at the germination stage according to the assessed traits (Appendix A; Appendix A). In the population, all the traits showed transgressive segregation with a wide variation (Figure 1). Under the 150 mM NaCl condition, the mean values of the LSRGI (relative germination index under low-salt conditions), LSRGR (relative germination rate under low-salt conditions), and LSRCL (relative coleoptile length under low-salt conditions) were 0.65, 0.79, and 0.65, with variation coefficients of 34.40%, 29.67%, and 36.37%, respectively. Under the 300 mM NaCl condition, the mean values of the HSRGI (relative germination index under high-salt conditions), HSRGR (relative germination rate under high-salt conditions), and HSRCL (relative coleoptile length under high-salt conditions) were 0.26, 0.50, and 0.45, with variation coefficients of 72.56%, 54.35%, and 36.78%, respectively. These results confirmed that salt tolerance at the rice germination stage is a typical quantitative trait. The Pearson correlation analysis demonstrated that there was a very significant positive correlation between RGI and RGR across salt stress; however, the RCL is not significantly correlated with RGI and RGR (Figure 2).

### 2.2. Linkage Mapping for Salinity Tolerance

The genetic basis associated with salt tolerance at the rice germination stage was deciphered using linkage mapping. With the ultra-high-density genetic map of JL1/MY23 RIL population, a total of twenty QTLs associated with salt-tolerant traits (LSRGR, LSRGI, LSRCL, HSRGR, HSRGI, and HSRCL) were detected on four chromosomes and explained 6.00%–16.00% of the phenotypic variation (Table 1).

Under low-salt stress conditions (150 mM NaCl), nine QTLs were identified. They were distributed on chromosomes 2, 3, and 7. Among them, two QTLs (*qLSRGI3-1* and *qLSRGI3-2*) on chromosome 3 explained more than 10% of the phenotypic variation. Under high-salt stress conditions (300 mM NaCl), eleven QTLs were identified, distributed on chromosomes 2, 3, 6, and 7 (Appendix A). Among them, the QTL *qHSRCL3-1*, whose contribution reached 16.0%, was identified as the major QTL and worthy of further exploration.

Significantly, six QTLs (*qLSRGR3-2*, *qLSRGI3-2*, *qLSRCL3-1*, *qHSRGR3-2*, *qHSRGI3-2*, and *qHSRCL3-1*) across two salt stress conditions, including major QTLs (contributing more than 15% phenotypic variation), were co-located in the physical region (Chr.3: 9200.47 Kb–11,961.23 Kb) and could be considered stable QTLs (Figure 3). Thus, given their value and reliability, this interval was strongly considered as an essential salinity tolerance region in rice for further exploring candidate genes.

According to annotation information from the Nipponbare reference genome (http://rice.plantbiology.msu.edu/, accessed on 11 January 2024), a total of 369 genes were distributed in the stable QTL region. Subsequently, the genes of differential expression were assessed to preliminarily determine the candidate genes in the stable and major QTL.

### 2.3. Gene Expression Profile and Comparative Transcriptome Analysis

To identify salt stress-responsive genes and complement the linkage analysis, differentially expressed genes (DEGs) between Jileng1 and Milyang23 under normal and salinity conditions were determined using RNA-seq. Under normal conditions, there are only 96 DEGs between parents, with 52 upregulated and 44 downregulated. In contrast, under salinity conditions, the DEGs between parents increased significantly to 3861, with 1510 upregulated and 2351 downregulated (Figure 4). Comparing the DEGs between different conditions for discovering salt-specific DEGs (described in Section 4.4), 3807 DEGs (2339 downregulated and 1468 upregulated) were specially detected under salinity conditions, and 41 DEGs (31 downregulated and 10 upregulated) were specially detected under normal conditions. There were 53 DEGs coincidently detected both under normal and salinity conditions, but 8 DEGs were also considered to be specific to salt stress, as they demonstrated a significantly different expression profile between normal and salinity conditions with |ΔLFC| > 1.5. In summary, a total of 3856 salt-specific DEGs were utilized for subsequent enrichment analysis and overlaid with stable QTLs (Appendix A).

A KEGG (Kyoto Encyclopedia of Genes and Genomes) pathway enrichment analysis was performed to characterize the biological roles of the salt stress-specific DEGs, and it revealed several important salt-related pathways (Appendix A). Metabolic pathways, including glutathione metabolism (osa00480), phenylpropanoid biosynthesis (osa00940), gibberellin biosynthesis (osa00904), ribosome (osa03010), starch and sucrose metabolism (osa00500), amino sugar and nucleotide sugar metabolism (osa00520), and plant–pathogen interaction (osa04626), were enriched (Figure 5).

### 2.4. Mining the Potential Candidate Genes in the Stable QTL Interval

To further narrow down the candidate genes, intersection analysis was conducted with salt-specific DEGs and all annotation genes within the stable QTL region (Chr3:9200.47 Kb–11,961.23 Kb). A total of 33 salt-specific DEGs were co-localized with the stable QTL. Furthermore, functional annotation of the 33 identified genes revealed that 5 genes were cloned, 8 genes had definite functional annotation, and 20 genes had an unclear function (Appendix A). According to the annotation information, eight genes or their homologs involved in salt stress regulation were regarded as putative candidate genes (Table 2).

To confirm the accuracy and reproducibility of DEGs identified using RNA-seq, the expression levels of these putative candidate genes were verified with qRT-PCR analysis, and the relative expression trends showed general consistency between the two methods (Figure 6). Therefore, the RNA-seq data were deemed reliable. Among them, the expression level of seven genes (*JIOsPR10*, *Os03g0279700*, *OsABCG5*, *OsGLYI3*, *OsGSTL2*, *OsCam1-1*, and *OsbZIP29*) showed significant difference under salt stress but non-significant difference under normal conditions. In contrast, the expression of *OsGSTL1* showed a significant difference under normal conditions but a non-significant difference under salt stress conditions.

### 2.5. Sequence Analysis of the Putative Candidate Genes

To further detect the variants and understand their roles associated with salt tolerance, the promoter and gene body of these putative candidate genes were analyzed using Sanger sequencing. As part of an aligning sequence in the gene body region, many mutations were detected in six genes (*Os03g0279700*, *OsABCG5*, *OsGSTL1*, *OsGSTL2*, *OsCam1-1* and *OsbZIP29*) (Appendix A). Nevertheless, only one non-synonymous mutation with a single base variant from A to C was detected on the second exon of *OsGSTL2*, which caused amino acid variation from threonine to arginine. All of the other nucleotide variations were located in a noncoding region or synonymous mutation.

In the promoter region, the alignment results revealed one, two, four, nine, and four sequence variations consisting of *OsGLYI3*, *Os03g0279700*, *OsABCG5*, *OsCam1-1*, and *OsbZIP29*, respectively. Subsequently, the cis-acting element was retrieved, and one notable Indel variation was found in *OsCam1-1*. In the salt-sensitive parents Jileng1, the promoter of *OsCam1-1* was consistent with the reference sequence (Nipponbare), while in the salt-tolerant parent Milyang23, there was a 10 bp insertion (ACATGATTGA) at the −1205 position from the initial transcription site, which contains the CAAT box element (Figure 7A; Appendix A). With the significant difference in gene expression levels of *OsCam1-1* between the two parents, we speculated that the variation in the promoter region, especially the transcription factor binding region (CAAT box), is the cause of the differential expression with *OsCam1-1*.

Haplotype analysis referring to this Indel sequence was conducted on a natural population, which was used to perform genome-wide association studies for relative germination index and relative germination rate under salinity conditions in our previous study [37], and it could divide the germplasm into two haplotypes. The cultivars carrying Hap2 which contained the insertion demonstrated inferior salt tolerance ability, as their relative germination index and relative germination rate were lower than cultivars with Hap1 (Figure 7(B-1,B-2), Appendix A).

## 3. Discussion

With the increasing promotion of direct seeding cultivation of rice, it is important to explore more loci or genes for seed germination under salt stress [38]. Breeders have made substantial efforts to understand the genetic mechanism, and more than 100 salt-tolerant QTLs at the seed germination stage have been identified using linkage mapping or GWAS methods [39]; nevertheless, there are still some deficiencies due to the low throughput of molecular markers [40]. In the present study, an RIL population with an ultra-high-density genetic map constructed with millions of SNP markers developed from genomic resequencing [41] was employed to discover the salt-tolerant QTLs.

Under two salt conditions, a total of twenty QTLs controlling seed germination-related traits were discovered, concentrated in four physical intervals (Chr2:10,199.59 Kb–19,240.04 Kb, Chr3:5638.24 Kb–8127.61 Kb, Chr3:9200.47 Kb–11,961.23 Kb, and Chr7:374.59 Kb–8127.61 Kb). In particular, the Chr3:5638.24 Kb–8127.61 Kb interval was noticeable, as six QTLs (*qLSRGR3-2*, *qLSRGI3-2*, *qLSRCL3-1*, *qHSRGR3-2*, *qHSRGI3-2*, and *qHSRCL3-1*) were co-located in this physical region and included the major QTL *qLSRCL3-1*, which contributed more than 15% phenotypic variation. Furthermore, many QTLs related to salt tolerance detected in previous studies partially or completely overlapped with this chromosomal region. Specifically, two QTLs (*qSTR-3a* and *qDM-3*) [42], six QTLs (controlling STS, CHL, and SSNP under salt stress) [43] and four QTLs (*qRTL3.10*, *qSRR3.10*, *qSRR3.11*, and *qSRR3.9*) [44] overlapped with this physical interval. The mutual comparison indicated that these QTLs are consistent across different genetic backgrounds, and it also verified their reliability. Thus, it was reasonably surmised that the salt-tolerant genes existed in this stable and major QTL region. However, it is still difficult to pinpoint the crucial salt-tolerant causal gene, as hundreds of genes were underlying this interval.

The developments in multi-omics technology have complemented the rapid discovery of genes specifically involved in salt tolerance [45]. In this study, a total of 3856 salt-specific DEGs were identified, enriched in seven KEGG pathways. Some of these pathways, such as glutathione metabolism, were confirmed to be closely associated with salt stress [46]. High glutathione (GSH) levels have often been correlated with a higher salt tolerance, and increasing GSH levels have been found to increase salt stress tolerance by reducing salt-dependent oxidative stress [47]. The function of these pathways in seed germination under salt stress merits in-depth study.

Meanwhile, incorporating RNA-seq into linkage analysis has the potential to meet the demand for prioritizing candidate genes [26]. According to intersection analysis, the number of putative candidates markedly decreased to 33 in Chr3:9200.47 Kb–11,961.23 Kb. Subsequently, functional annotation revealed the potential role of eight genes involved in salt stress regulation. Among the putative candidates, four genes were cloned and functionally analyzed, and the other four genes’ homologs were found to be involved in salt stress regulation.

To further decipher their roles associated with salt tolerance, the promoter and gene body of the putative candidate genes were analyzed using Sanger sequencing, which can remedy the limitation of random error and indiscoverable long insertion and deletion variation experienced when using de novo genomic sequencing [48]. Notably, one Indel (10 bp, ACATGATTGA) was discovered in *OsCam1-1*, which is located at −1205 bp from the initial transcription site and contains the cis-acting element of the CAAT box. The CAAT box is known as the basal and core promoter, which initiates transcription of a particular gene [49]. We speculate that this Indel may be the inducement of transcriptional expression difference in *OsCam1-1*, and it was verified using the results of gene expression and haplotype analysis in this study (Figure 6).

*OsCam1-1* is a salt stress-responsive calmodulin and has been shown to play an important role in the signal transduction cascade in proline accumulation during salt stress [50]. *OsCam1-1* showed higher expression in a sensitive parent (Miyang23) than a tolerant parent (Jileng1) under salt stress (Figure 6), suggesting that it may play a negative role in rice salt tolerance. This was further confirmed with haplotype analysis, as the cultivars with insertion sequence containing the CAAT box demonstrated inferior salt-tolerant ability (Figure 7). The deduction is not inconsistent with the results of Kaewneramit et al. [35], which indicated that *OsCaM1-1* overexpression in the transgenic rice mitigated salt-induced oxidative damage. The dissimilarity may be caused by the difference in genetic background and growth stage with the experimental material. In the future, further experimentation is required to validate the function of *OsCam1-1*. The KEGG enrichment analysis revealed that *OsCam1-1* is involved in the significant enrichment pathway (osa04626, plant–pathogen interaction), which includes 22 calmodulin genes related to calcium signal transduction. Their roles and regulatory relationship in the salt stress response of rice should be validated with more molecular biology experiments.

## 4. Materials and Methods

### 4.1. Experimental Site and Plant Material

The present study was conducted at the salt identification base of HAAFS, Tangshan, China (39°20′ N, 118°17′ E), in 2023. An RIL population containing 253 lines, derived from a Jileng1 (salt-tolerant at germination stage) and Milyang23 (salt-sensitive at germination stage) cross [41], was used to detect seed germination ability under salt stress. The salinity tolerance phenotypes of Jileng1 and Milyang23 are shown in Appendix A.

### 4.2. Evaluation of Salt Tolerance at Seed Germination Stage

The seed germination experiment was carried out under three treatments (H_2_O, 150 mM NaCl, 300 mM NaCl). Every independent treatment was in a completely randomized design (CRD) with three replications. Seeds from 253 lines, along with their parents, were incubated at 50 °C for 5 days to eliminate the effect of dormancy on germination. All of the healthy and plump grains were disinfected with 1.4% NaClO for 15 min and then rinsed three times with sterile distilled water. Thirty seeds per line were placed in Petri dishes (d = 9 cm) with two layers of filter paper, to which 15 mL of target solution or distilled water was added. Subsequently, all Petri dishes were placed in a growth chamber with a temperature of 30 °C and a 12 h light (200 μmol m^−2^ s^−1^)/12 h dark photo-period for 10 days. The solution was replaced every 2 days to maintain the NaCl concentration and the distilled water volume.

The germination rate (GR), germination index (GI), and coleoptile length (CL) were investigated for RILs and their two parents under three treatments. The seeds with germinability were observed each day to calculate the germination rate (GR) and germination index (GI) according to the method of Ju et al. [37]. GI = Σ(Gt/Tt), where Gt is the number of germinated seeds on day t, and Tt is the time corresponding to Gt in days. Ten days later, 10 seedlings were selected in each dish, and their coleoptile length was measured.

The six germination-derived traits LSRGR (relative germination rate under low-salt conditions), LSRGI (relative germination index under low-salt conditions), LSRCL (relative coleoptile length under low-salt conditions), HSRGR (relative germination rate under high-salt conditions), HSRGI (relative germination index under high-salt conditions), and HSRCL (relative coleoptile length under high-salt conditions) were further calculated using the following equation to assess the salt tolerance of RILs.

Relative value = (phenotype value under salt stress)/(phenotype value under control conditions).

### 4.3. Linkage Mapping

A phenotypic frequency distribution map of six traits was generated with the “ggplot2” package (version 3.5.1). The multi-interval mapping function (MQM) was employed to detect QTLs for six traits with MapQTL6.0 [51]. The LOD threshold was determined by applying 1000 permutation tests with 5% probability. The physical confidence interval of the QTL corresponded to a 1.5-LOD decrease from the peak LOD value, and the QTLs with overlapping intervals were merged. QTLs detected across two salt conditions and related traits were considered to be stable QTLs and were used to carry out subsequent candidate gene prediction. Genes located in stable QTL intervals were annotated based on Ensembl (http://rice.plantbiology.msu.edu/, accessed on 11 January 2024) databases.

### 4.4. RNA Sequencing and Data Analyses

The seeds of Jileng1 and Milyang23 under 150 mM NaCl and H_2_O treatment (3 repetitions) for 3 d were sampled and immediately frozen in liquid nitrogen for RNA extraction. The total RNA was extracted using a Sigma Spectrum Plant Total RNA Kit (Sigma-Aldrich, St. Louis, MO, USA). Then, 4 mg of total RNA with RIN ≥ 8 (Bioanalyzer 2100, Agilent Technologies, Santa Clara, CA, USA) was prepared for library construction and sequenced on the HiSeq2500 platform, according to the standard procedures of Novogene Bioinformatics Technology Co., Ltd., Tianjin, China. The raw sequencing datasets generated for the current study are available in the Genome Sequence Archive, https://ngdc.cncb.ac.cn/gsa (accessed on 17 July 2024) (accession no. CRA017783).

The raw RNA-seq data of 12 samples (two treatment × two parents × three repetitions) were filtered using fastp [52] and mapped to the Nipponbare genome (http://plants.ensembl.org/Oryza_sativa/, accessed on 11 January 2024). Read counts were generated using HISAT [53] with default parameters and normalized using Deseq2 with the TMM method [54]. Normalized read counts were utilized for exploring differentially expressed genes (DEGs) between Jileng1 and Milyang23 under salt stress and normal conditions.

DEGs were discriminated according to |log2 fold change| ≥ 1.5 and a false discovery rate (FDR) of ≤0.05. Subsequently, the salt-specific DEGs were determined by comparing between (S1 vs. S2) and (N1 vs. N2), and unshared DEGs were regarded as salt-specific DEGs. Therefore, there were six classes of unshared DEGs that were considered: genes that were upregulated (i, “SalOnly_up”) or downregulated (ii, “SalOnly_down”) only in the salt stress conditions and unchanged in the normal conditions; genes that were upregulated (iii, “NorOnly_up”) or downregulated (iv, “NorOnly_down”) only in normal conditions and unchanged in salt stress conditions; and DEGs that were detected under both salt stress and normal conditions but showed a difference in LFC (ΔLFC = LFC of the salt stress condition −LFC of the normal condition), (v, “ΔLFC > 1.5”) or (vi, ΔLFC < −1.5). The types i, iv, and v were regarded as upregulated DEGs specifically related to salt stress, and the types ii, iii, and vi were considered downregulated DEGs specifically related to salt stress. The types SalOnly_up, NorOnly_down, and ΔLFC > 1.5 were regarded as upregulated DEGs specifically related to salt stress, and the types SalOnly_down, NorOnly_up, and ΔLFC < −1.5 were considered downregulated DEGs specifically related to salt stress. A KEGG pathway enrichment analysis of salt-specific DEGs was performed using the package ClusterProfiler [55], and pathways with an adjusted *p*-value of <0.05 were considered to be significantly enriched.

### 4.5. Validation of Candidate Genes Using Quantitative Real-Time PCR and Sequence Analysis

Eight genes from the list of possible candidate genes were validated using qRT-PCR and sequence analysis (Appendix A). Three independent biological replicates for each sample were performed for qRT-PCR using a Roche LightCycler96 (Munich, Germany). The PCR cycles included 2 min at 95 °C, followed by 40 cycles of 5 s at 95 °C and 30 s at 60 °C. The actin gene of rice (*Os04g0177600*) was employed as the internal control. Transcript levels of nominated genes from three biological replicates were computed as 2^−ΔΔCt^.

Seed coleoptile samples were collected from two parents. DNA was extracted using the CTAB method and quantified using both a NanoDrop ND-1000 Spectrophotometer (Wilmington, DE, USA) and agarose gel electrophoresis. The gene body and 1.5 Kb promoter region sequence of Jileng1 and Milyang23 with eight candidate genes were cloned using PCR-based sequencing. To characterize the sequence variation of the potential candidate gene (*OsCam1-1*), PCR-based sequencing on the segment of promoter region was analyzed in a natural population, comprising 162 accessions of cultivated rice (Appendix A). Sequence alignments and variation exploration were performed using DNAMAN10.0 (Lynnon Biosoft, https://www.lynnon.com/).

## 5. Conclusions

In the present study, the strategy of integrating linkage mapping, comparative transcriptome analysis, and gene sequencing was employed to investigate the genetic basis and excavate target genes related to salt tolerance in rice. A major and stable QTL, *qRCL3-1*, associated with relative length and located at the Chr3:9200.47 Kb–11,961.23 Kb interval, was identified and highlighted. Due to a comparison of the transcriptome profile complemented with gene annotation, eight putative candidate genes were further identified. Based on the sequence variation between two parents, *OsCam1-1* was confirmed to be the potential gene controlling salt tolerance in rice. This study provides insights for further manipulating the essential mechanism of salt tolerance in rice, and related molecular markers of *OsCam1-1* will provide a tool for molecular breeding programs in rice and facilitate improving salt tolerance in the future.

## Figures and Tables

**Figure 1 ijms-25-10376-f001:**
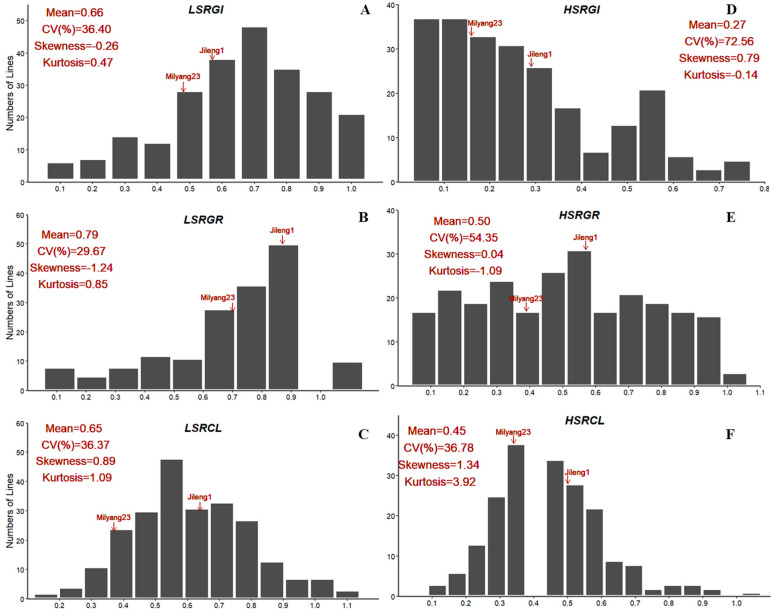
Phenotypic distribution of six salt tolerance-related traits at germination stage: (**A**) LSRGI; (**B**) LSRGR; (**C**) LSRCL; (**D**) HSRGI; (**E**) HSRGR; (**F**) HSRCL.

**Figure 2 ijms-25-10376-f002:**
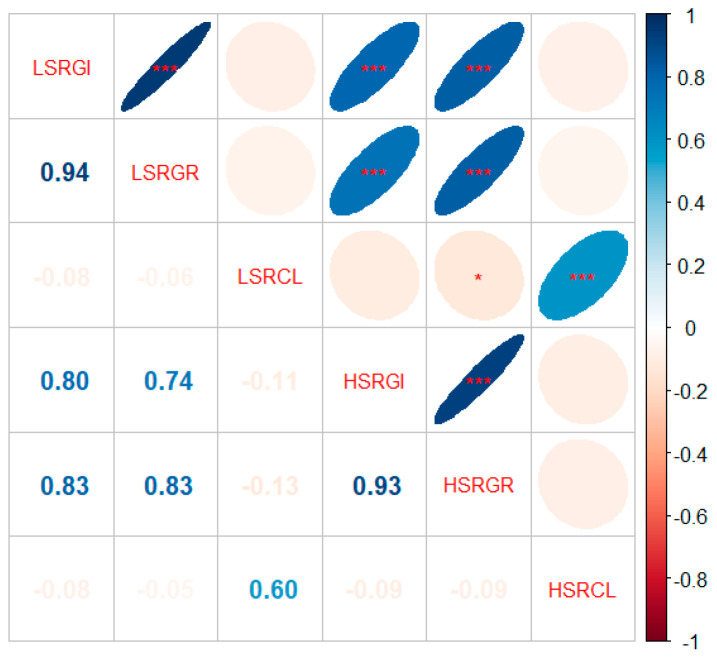
Pearson correlation analysis between six salt tolerance-related traits (LSRGI, LSRGR, LSRCL, HSRGI, HSRGR, HSRCL). The lower part is the correlation coefficient between traits, and the upper part is the significant testing label. * and *** indicate significance level at 0.05 and 0.001, respectively.

**Figure 3 ijms-25-10376-f003:**
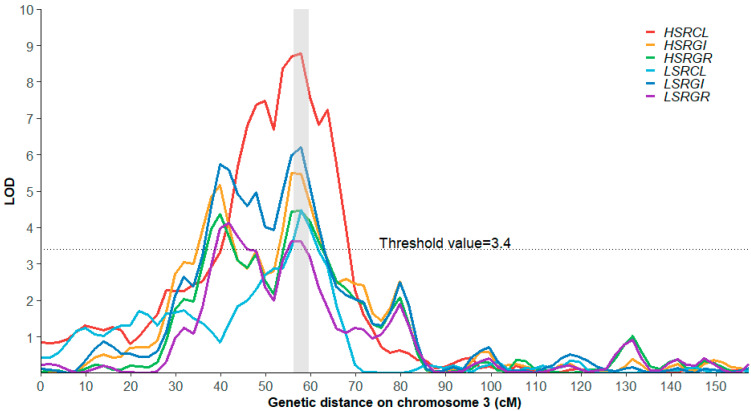
QTL mapping with six salt tolerance-related traits on chromosome 3. The *x*-axis shows the genetic distance on chromosome 3 with unit cm, while the *y*-axis represents the LOD scores. The gray span is the confidence interval (genetic distance of Chr3:52.284–62.177 cm, and corresponding physical interval Chr3:9200.47–11,961.24 Kb) of the stable and major QTL.

**Figure 4 ijms-25-10376-f004:**
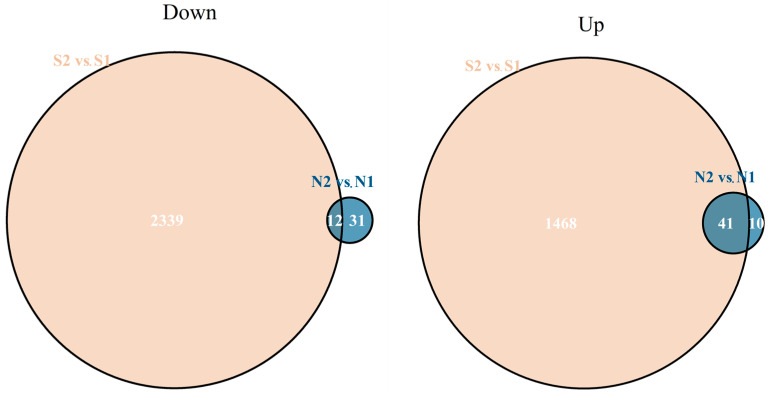
Venn diagram of DEGs between S2 vs. S1 and N2 vs. N1. S2, S1, N2, N1 represent Milyang23 under salt stress, Jileng1 under salt stress, Miyang23 under H_2_O, Jileng1 under H_2_O conditions, respectively.

**Figure 5 ijms-25-10376-f005:**
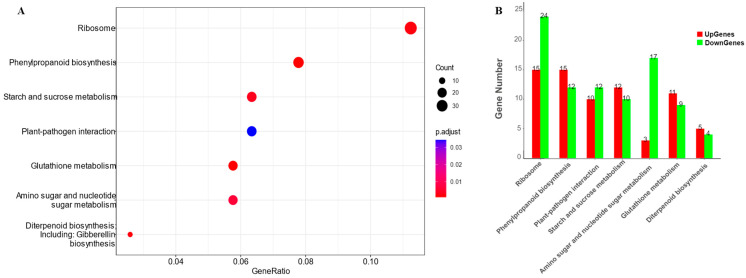
The KEGG pathways enriched for the salt stress-specific DEGs: (**A**) enriched KEGG pathways demonstrated in a dot plot; (**B**) numbers of up- and downregulated genes with parents under salt stress enriched in KEGG pathways.

**Figure 6 ijms-25-10376-f006:**
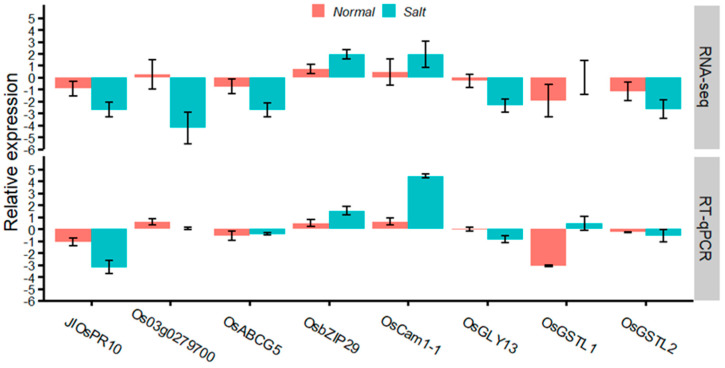
Gene expression changes under salt stress and normal conditions based on qRT-PCR and RNA-seq.

**Figure 7 ijms-25-10376-f007:**
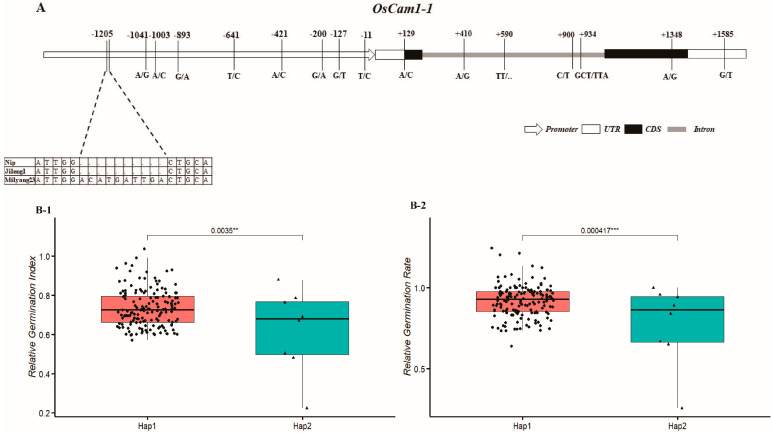
Sequencing and haplotype analysis of *OsCam1-1*: (**A**) variants located on the promoter and gene body of *OsCam1-1*; (**B-1**) haplotype analysis on Indel with relative germination index under salinity conditions; (**B-2**) haplotype analysis on Indel with relative germination rate under salinity conditions. ** and *** indicate significance level at 0.01 and 0.001, respectively.

**Table 1 ijms-25-10376-t001:** QTLs associated with rice salt-tolerant traits at the germination stage.

Trait	QTL	LOD	Chromosome	Genetic Interval (cM)	Physical Interval (Kb)	PVE (%)	Additive
LSRGI	*qLSRGI2-1*	3.65	2	48.80–62.33	11,133.44–19,240.04	6.40	0.06
	*qLSRGI3-1* *	6.48	3	38.49–45.23	6112.64–6965.22	11.10	0.08
	*qLSRGI3-2*	6.88	3	54.76–59.62	9618.18–11,348.70	11.80	0.09
	*qLSRGI7-1*	3.55	7	10.10–14.25	2208.42–5052.83	6.30	0.06
LSRGR	*qLSRGR2-1*	3.61	2	45.55–48.80	10,199.59–11,133.44	6.40	0.06
	*qLSRGR3-1*	4.42	3	36.20–48.97	6039.23–8127.61	7.70	0.07
	*qLSRGR3-2*	3.42	3	52.28–62.18	9200.47–11,961.23	6.00	0.06
LSRCL	*qLSRCL3-1*	5.11	3	56.28–61.02	9799.94–11,603.07	8.90	0.07
	*qLSRCL7-1*	3.92	7	18.72–29.40	6046.91–8992.24	6.90	0.06
HSRGI	*qHSRGI2-1*	3.65	2	57.65–62.33	18,228.53–19,240.04	6.40	0.05
	*qHSRGI3-1*	5.39	3	35.03–42.21	5638.24–6928.16	9.30	0.06
	*qHSRGI3-2*	6.16	3	54.76–58.76	9618.18–10,640.21	10.60	0.07
	*qHSRGI6-1*	3.88	6	49.50–55.14	6938.15–8230.83	6.80	0.05
	*qHSRGI7-1*	3.63	7	12.91–25.40	374.59–2208.42	6.40	0.05
HSRGR	*qHSRGR2-1*	3.79	2	57.24–62.33	18,438.09–19,240.04	6.70	0.07
	*qHSRGR3-1*	4.63	3	36.20–45.23	6039.23–6965.22	8.10	0.08
	*qHSRGR3-2*	4.88	3	52.28–62.18	9200.47–11,961.23	8.50	0.08
	*qHSRGR6-1*	3.57	6	55.25–64.50	8036.25–10,246.51	6.30	0.07
	*qHSRGR7-1*	4.01	7	10.10–25.40	374.59–4817.93	7.00	0.07
HSRCL	*qHSRCL3-1*	9.55	3	56.28–58.51	9799.94–10,540.07	16.00	0.07

* indicates QTLs whose PVE (phenotypic variation explained) is more than 10%.

**Table 2 ijms-25-10376-t002:** The potential candidate genes in the stable QTL interval.

Gene ID	Name	Position (bp)	Annotation	Reference
*Os03g0277500*	*OsGLYI3*	3:9,418,945–9,419,670	A glyoxalase gene expressed in rice seed that contributes to seed longevity and salt stress tolerance	[29]
*Os03g0279700*		3:9,524,919–9,525,850	ZOS3-09-C_2_H_2_ zinc finger protein	[30]
*Os03g0281900*	*OsABCG5*	3:9,654,333–9,657,024	Rice stomatal closure requires guard cell plasma membrane ATP-binding cassette transporter	[31]
*Os03g0283100*	*OsGSTL2*	3:9,712,224–9,714,920	Glutathione S-transferase gene	[32]
*Os03g0283200*	*OsGSTL1*	3:9,717,253–9,719,884	Glutathione S-transferase gene	[33]
*Os03g0300400*	*JIOsPR10*	3:10,562,231–10,563,277	Jasmonate inducible pathogenesis-related class 10 protein gene	[34]
*Os03g0319300*	*OsCam1-1*	3:11,522,676–11,524,356	Salt stress-responsive calmodulin	[35]
*Os03g0322700*	*OsbZIP29*	3:11,683,525–11,684,400	Basic leucine zipper transcription factor	[36]

## Data Availability

Data are contained within the article.

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
