# Peer review of "Integration Linkage Mapping and Comparative Transcriptome Analysis to Dissect the Genetic Basis of Rice Salt Tolerance Associated with the Germination Stage"

_ijms, 2024, doi:10.3390/ijms251910376_

Round 1
Reviewer 1 Report
Comments and Suggestions for Authors
Please see attached for comments and questions.

Author Response
Dear editors and reviewers,
Thank you very much for the encouraging comments. Those comments are all valuable and very helpful for revising and improving our paper. We had studied the comments carefully and have made correction which we hope meet with approval.
Major concerns
1.Please upload the supplementary file for review.
We have already uploaded the supplementary file in the new submition.
- For all the abbreviations used, please present them in full name when first used in the manuscript.
We had checked the abbreviations and supplement full name.
Minor comments
1. Please enlarge the texts in Figure 1 to make them more readable.
We already rearranged and enlarged the Figure 1.
- Line 89, what is the statistical test to support that the traits showed normal distribution?
We didn’t carry out statistical test for normal distribution, only infer the data distribution by kurtosis and skewness. Some of the data is not normal distribution, and we had already omitted this sentence.
- Line 93, what are the p-values to support ‘significant down trend’?
We didn’t carry out significance test, and just comparing the value of LSRGI (Relative germination index under low salt condition), LSRGR (Relative germination rate under low salt condition) and LSRCL (Relative coleoptile length under low salt condition) with HSRGI (Relative germination index under high salt condition), HSRGR (Relative germination rate under high salt condition) and HSRCL (Relative coleoptile length under high salt condition).
- Figure 2, please describe the data used and the correlation test (pearson or spearman) used to perform the correlation analysis.
We perform pearson correlation test between six traits.
- Figure 3, what is the unit of the x-axis? Is it Kb?
In figure3, we used genetic distance cM as scales in x-axis.
6. Figure 4, please be more specific in the text regarding how ’44 DEGS’ and ‘3856 salt-specific DEGs’ are derived? Are there only 8 genes with log fold change>1.5? What is the log base here?
We had already specified and correct those result. The 3807 DEGs (2339 down-regulated and 1468 up-regulated) were specially detected under salinity condition, and 41 DEGs (31 down-regulated and 10 up- regulated) were specially detected under normal condition. There were 53 DEGs common detected both under normal and salinity conditions, but eight DEGs shown |â–³LFC|>1.5, were also considered to be specific to salt stress.
- Please enlarge the texts in Figure 5 to make them more readable.
We had already change Figure 5B to barchart with significant KEGG pathway, which it is more readable.
8. Figure 6, it seems the expression levels of OsABCG5 are not consistent between RNA-seq and RT-qPCR.
The relative express level of OsABCG5 is significant downregulated, detected by RNA-seq, and not significant by qRT-PCR. But the trend between two method is general consistency. As experiment error, the result is always not exactly consistent between two methods.
- For line 176-180, where can I get such information and how can these observations
lead to the following conclusion?
Thanks for your recommending. We had already deleted this conclusion which is not strong supported by result.
- Figure 7B, how were the germination index and rate derived or achieved?
The germination index and rate were achieved in our previous study, which was used to perform genome-wide association studies for relative germination index and relative germination rate under salinity condition
Ju, C.Y.; Ma X.D.; Han, B.; Zhang, W.; Zhao, Z.W.; Geng L.Y.; Cui D.; Han, L.Z. Candidate gene discovery for salt tolerance in rice (Oryza sativa L.) at the germination stage based on genome-wide association study. Front. Plant Sci. 2023, 13:1010654. http://doi.org/ 10.3389/fpls.2022.1010654
Reviewer 2 Report
Comments and Suggestions for Authors
Dear authors,
You have chosen to research a very interesting topic. From a scientific point of view, salt tolerance of rice at the germination stage is one of the major determinants for the stable stand establishment, which is very important for direct seeding in salinity soil. On one side your study will provide important information for elucidating the genetic and molecular basis of rice salt tolerance at the germination stage, and on the other side, the detected in your research genes will be useful for the improvement of rice salt tolerance.
I have no significant remarks, but I still want to make some clarifications that would increase the quality of your work.
1. In the results (line 89) you claim that all the traits shown in Figure 1 have a normal or nearly normal distribution, which is not correct. The traits in Figure 1 E and D have different distributions. On line 94 you wrote that these traits have very big coefficients of variation (70.71%, 54.35%). In my opinion, they should be omitted from the next points of the analysis.
2. In the article you discuss the results from Supplementary Materials (Figures S1, S2, Tables S1, S2, S3, etc.), but I couldn’t find some file with these results. I take your reported results on trust, without being able to comment on them. It was very difficult for me to navigate the results, which automatically leads me to think that the reader will encounter the same difficulty.
3. I advise you to include a brief description of the abbreviations used in the legends under the tables, even if you have already included them in the text (e.g., PVE in Table 1). This would improve the quality of your presented results and make it easier for the reader to understand them.
4. The results in Figure 5B are very small and difficult to read. It would be good to revise this figure so the results become visible and readable.
Overall, I am impressed by the thoroughness of your research and believe it will have an impact on the scientific community.
Comments on the Quality of English Language
The English language used in the manuscript is good, but some small grammatical and punctuation errors can be a bit distracting for the reader. My advice to the authors is to consider using the services of a professional translator, which could significantly enhance the quality of the presented material.
Author Response
Dear editors and reviewers,
Thank you very much for the encouraging comments. Those comments are all valuable and very helpful for revising and improving our paper. We had studied the comments carefully and have made correction which we hope meet with approval.
1.In the results (line 89) you claim that all the traits shown in Figure 1 have a normal or nearly normal distribution, which is not correct. The traits in Figure 1 E and D have different distributions. On line 94 you wrote that these traits have very big coefficients of variation (70.71%, 54.35%). In my opinion, they should be omitted from the next points of the analysis.
Thanks for your advisements. We had already correct the improper conclusion.
- In the article you discuss the results from Supplementary Materials (Figures S1, S2, Tables S1, S2, S3, etc.), but I couldn’t find some file with these results. I take your reported results on trust, without being able to comment on them. It was very difficult for me to navigate the results, which automatically leads me to think that the reader will encounter the same difficulty.
We have already uploaded the supplementary file in the new submition.
- I advise you to include a brief description of the abbreviations used in the legends under the tables, even if you have already included them in the text (e.g., PVE in Table 1). This would improve the quality of your presented results and make it easier for the reader to understand them.
Thanks for your advisements. We had supplemented full name in table.
- The results in Figure 5B are very small and difficult to read. It would be good to revise this figure so the results become visible and readable.
We had already change Figure 5B to barchart with significant KEGG pathway, which it is more readable.